# Protocol for the OCAY study: a cohort study of orphanhood and caregiver loss in the COVID-19 era to explore the impact on children and adolescents

Kathryn J Steventon Roberts [1,2] Stefani Du Toit,[3] Tatenda Mawoyo,[3] Mark Tomlinson [3,4] Lucie D Cluver,[1,5] Sarah Skeen,[3,6] Christina A Laurenzi,[3] Lorraine Sherr [2]

KJSR, SDT and TM contributed equally.

For numbered affiliations see end of article.

**Correspondence to**
Professor Lorraine Sherr;
l.sherr@ucl.ac.uk

## ABSTRACT

**Introduction** Globally, no person has been untouched by the COVID-19 pandemic. Yet, little attention has been given to children and adolescents in policy, provision and services. Moreover, there is a dearth of knowledge regarding the impact of COVID-19-associated orphanhood and caregiver loss on children. This study aims to provide early insights into the mental health and well-being of children and adolescents experiencing orphanhood or caregiver loss in South Africa.

**Methods and analysis** Data will be drawn from a quantitative longitudinal study in Cape Town, South Africa. A sample of children and adolescents between the ages of 9 and 18 years, experiencing parental or caregiver loss from COVID-19, will be recruited together with a comparison group of children in similar environments who did not experience loss. The study aims to recruit 500 children in both groups. Mental health and well-being among children will be explored through the use of validated and study-specific measures. Participants will be interviewed at two time points, with follow-up data being collected 12–18 months after baseline. A combination of analytical techniques (including descriptive statistics, regression modelling and structural equation modelling) will be used to understand the experience and inform future policy and service provision.

**Ethics and dissemination** This study received ethical approval from the Health Research Ethics Committee at Stellenbosch University (N 22/04/040). Results will be disseminated via academic and policy publications, as well as national and international presentations including high-level meetings with technical experts. Findings will also be disseminated at a community level via various platforms.

## STRENGTHS AND LIMITATIONS OF THIS STUDY

⇒ This is the first empirical investigation of the impacts of COVID-19-associated caregiver loss on the well-being of children and youth.
⇒ A key strength of the study is the broad array of domains covered by the questionnaire used within this study and, the use of validated and standardised scales.
⇒ The longitudinal nature of the study will strengthen investigations relating to the direction of any associations identified.
⇒ At this stage, data will only be drawn from the Western Cape province of South Africa, potentially restricting the generalisability of results beyond the region.

Yet, there has been little attention focused on the children and adolescents experiencing the death of parents and caregivers and the resulting consequences for their health and well-being. Orphanhood is defined as the death of one or both parents.[3] Orphanhood has previously been associated with an increased risk of poor mental health, violence and abuse exposure, unstable home environments, school disruption, early sexual debut and poverty.[4–11] In many parts of the world, alternative caregivers, such as grandparents, are also key in providing essential care for children and adolescents. Due to the multigenerational nature of households, it is also important to consider caregiver loss as grandparents are often primary caregivers and were the most vulnerable to morbidity and mortality due to COVID-19.[1 2 12–14]

There needs to be a clear understanding and tracking of the nature of orphanhood and caregiver loss so that supportive provisions for children can be accurately tailored.[15] The occurrence of COVID-19 orphanhood and caregiver loss may well be layered onto

## BACKGROUND

The COVID-19 pandemic has had a dramatic impact on people across the world. The mortality rate from the virus has been high, with no country untouched, and notable deaths among parents and caregivers of children and adolescents—an estimated 10.5 million children and adolescents have experienced the death of a parent or caregiver.[1 2]

pre-existing social systems where children already experience a multitude of social challenges and other forms of loss, for example, due to HIV or tuberculosis.[16] However, there may be elements of orphanhood and caregiver loss associated with COVID-19 that are new and yet undescribed and uncharted. Similarities from previous large-scale parental loss experiences such as HIV/AIDS, Ebola, war and disaster needs to be understood while this new group of children and youth need to be linked to existing provision.

Undertaken within South Africa, the Orphanhood and Caregiver loss among Adolescents and Youth in the COVID-19 era (OCAY) study is to the best of our knowledge, the first empirical study to explore the impact of orphanhood and caregiver loss on the well-being of children and adolescents. Children and adolescents who have experienced the loss of a caregiver due to COVID-19 and a comparison group of children and adolescents who have not experienced loss will be recruited to the study. There are five core objectives of the study: (1) to determine if mental health outcomes (positive/negative) differ between children and adolescents who have experienced a COVID-19-associated caregiver death, and children and adolescents who have not experienced a COVID-19-associated caregiver death; (2) to explore the impact of COVID-19-associated caregiver death on health and well-being outcomes of children, and service use (comparing those with and without a caregiver loss, and exploring impacts in relation sociodemographic factors, eg, experience of poverty); (3) to explore how mental health outcomes (positive/negative) among children and adolescents who have experienced orphanhood and caregiver loss change over time (using follow-up data); (4) to examine associations between access to services (such as COVID-related grants, bereavement referrals) and outcomes for children and adolescents; (5) to compare the health and well-being status of children and adolescents who have lost a caregiver due to COVID-19 with children and adolescents who have lost a caregiver due to other causes (eg, HIV/AIDS). Objectives will use both cross-sectional and longitudinal data to explore changes in experience over time. Based on previous literature, it is hypothesised that children and adolescents experiencing COVID-19-associated caregiver loss will experience worse health, well-being and service use outcomes compared with children who have not experienced caregiver loss.

## METHODS AND ANALYSIS
### Study setting
The study will predominantly take place in Khayelitsha, which is a under-resourced peri-urban settlement on the outskirts of Cape Town, South Africa. Khayelitsha is the third largest township in South Africa, with an estimated population of between 400 000 and 750 000.[17] The majority of residents live in overcrowded informal housing with shared communal water taps and inadequate sanitation arrangements.[17] According to the 2011 Census, the average annual household income was R20 000 (approximately US$1200).[18] There is a high unemployment rate in Khayelitsha, approximately 50%.[19] Khayelitsha experiences high levels of crime with inconsistent policing.[20] As existing studies forms part of our recruitment approach, participants residing in other parts of South Africa will also be considered for inclusion.

### Participant recruitment
The study seeks to identify at least 500 children and adolescents aged between 9 and 18 years, who have been orphaned or lost a caregiver due to COVID-19 (defined as COVID-19 infection or likely COVID-19 infection). A similar proportion (n=500) of children and adolescents (9–18 years of age) who have not lost a parent or caregiver due to COVID-19 will be recruited as a comparison group. Various degrees of loss will be explored in the study (ie, parental loss, other caregiver loss, no loss). To identify the sample, a mosaic of five strategies will be used: (1) *existing studies*—as a starting point, participants included in previous studies conducted by the group's primary investigators and co-primary investigators will be contacted to assess eligibility.[21–25] These studies include populations similar to the population of interest (eg, age, sociodemographic characteristics) and are likely to include orphanhood-affected and non-affected participants (who have previously given consent to be contacted for future studies); (2) *community links*—neighbourhoods within our study setting often have street committees, and these committees will be used to identify eligible participants. Alternatively, identification could take place by door-to-door recruitment; (3) *schools*—following permission from the Western Cape Education Department, primary and high schools within the area will be approached for recruitment; (4) *non-governmental organisations (NGOs)*—NGOs within our study setting, rendering services to children and adolescents, will be used; (5) *snowball sampling*—participants recruited into the study will be provided with the opportunity to refer other potential participants.

### Patient and public involvement
It was not appropriate or possible to involve patients or the public in the design, or conduct, or reporting, or dissemination plans of our research at the time of writing this protocol.

### Study measures
All participants will be interviewed at two time points—baseline and 12–18 months later. Baseline data collection will take place between July 2022 and May 2023, with a second wave of data collection scheduled between July 2023 and May 2024. A study-specific inventory, adjusted to be age-appropriate, will be constructed. The components will use validated measures which have been used in five previous studies undertaken with children and adolescents within South Africa (the Young Carers study, the Mzansti Wakho study, the HEY BABY study, the Child

Community Care Study and the Adolescent Bereavement Initiative). Given the broad extent of the impact of COVID-19 and orphanhood on the lives of the children and adolescents, measures (for both those affected by orphanhood, and the non-affected group) will explore multiple aspects of life including: (1) demographic and social circumstance factors; (2) mental health (including depression, anxiety, post-traumatic stress and suicidality) and well-being; (3) COVID-19 experience; (4) bereavement experience; (5) care and subsequent care experience; (6) experience of support and provision (7) health experience; (8) education and school experience; (9) behavioural risk; (10) violence exposure, stigma and bullying experience and (11) parenting and caregiving experience. Data relating to the period between caregiver loss and baseline data collection will also be obtained to ensure that time since loss is accounted for within analyses. Measures will be piloted and refined/adapted based on participant feedback. Piloting will be undertaken with 10–15 participants who have been identified as eligible for the study within the greater Khayelitsha area. Study measures will remain similar across both waves, however minor amendments may be made based on feedback (eg, from data collectors and/or the study team) from baseline data collection. Measures that remain static over time will not be included within follow-up data collection. See table 1 for an overview of measures to be used within the study.

## Power calculation

Power calculations were based on the core mental health measures within the study (depressive/anxiety symptoms). Multiple power calculations have informed this estimate. Previous studies identifying approximately 10% of children and adolescents experience poor mental health.[26] Stratification modelling data based on the methodologies used in the study by Hillis et al,[2] identified that caregiver loss within South Africa due to COVID-19 has impacted approximately 211 000 children (estimate obtained November 2022; COVID-19 Orphanhood Calculator). Based on these data, a confidence level of 95% and a 2% margin of error, the sample size required to detect meaningful difference in the proportion of poor mental health symptomology reported between the sample and general population estimates ($\chi^2$ test) is 861. A second power calculation was undertaken to account for group differences (COVID-19 caregiver loss vs no COVID-19 caregiver loss) in the core mental health measures (prevalence). Based on identifying a prevalence of 5% poor mental health symptoms (depression/anxiety) among the control group (no COVID-19-associated orphanhood) and 10% prevalence within the population of interest (COVID-19-associated orphanhood), an alpha of 0.05, and 90% power the number of participants required to detect a meaningful difference between groups is 671 (COVID-19-associated orphanhood) vs 503 (no

COVID-19-associated orphanhood). Based on these estimates, and to account for attrition in planned follow-up rounds of data collection, a planned sample of 1000 children including the number of children and adolescents experiencing COVID-19-associated orphanhood or caregiver loss n=500, and the comparison group of children and adolescents experiencing no COVID-19-associated orphanhood n=500.

## Data collection procedures

Data will be collected by experienced, trained and supervised data collectors. Data collectors, with extensive prior experience in quantitative research in the study site, will undergo a 2-week refresher training workshop in general principles of research ethics, referral procedures, management of difficult situations, obtaining informed consent and how to administer the questionnaire. Data collectors will be fluent in English and isiXhosa (the vernacular language of Khayelitsha residents), with prior experience working with vulnerable youth and families in these settings. Assessments will take place at a research centre in Khayelitsha. In the instances where participants are unable to travel to the research centre, assessments will be conducted in a private space at the participant's home, school or NGO. In the instances where participants reside in other parts of South Africa, we will conduct telephonic assessments. Participants will be provided with transportation to and from the venue where assessments are taking place. Prior to starting the interview, caregivers of children under the age of 18 years will provide written consent for the participation of their child, and the child will provide informed assent. Participants who are 18 years of age will provide their own informed consent. All measures will be translated into isiXhosa and back-translated into English to ensure the use of accurate and quality translations. Interviews are expected to last between 60 and 90 min. During assessment, participants will be provided with breaks during which they will be served refreshments. All participants are provided with a reimbursement for their time, inconvenience and expense in the form of a shopping voucher, and these reimbursements will be provided at each time point. As a means of quality control, the assessments will be audio-recorded, with the permission of the participant. These audio-recordings are reviewed by supervisory staff on a regular basis to ensure consistency in administration between data collectors. On completion of the assessments, participants will be provided with a further opportunity to ask any questions or express any concerns.

The second wave of data collection will follow a similar format to the first wave of data collection. As with the first wave, data collectors will undergo a refresher training workshop. Participants will be recontacted by the study team. Retention will be a core priority within the second round of data collection. The study team are known to the local community. Participants will be contacted on numerous occasions by the data collection team. The core study team have an excellent track-record of high

**Table 1** Study measures for inclusion within the OCAY study

| Domain | Measures | Example items and response categories | Examples of previous use and/or validation within LMIC |
|---|---|---|---|
| **Demographic and social circumstance factors** | | | |
| Demographic information | Child age, biological sex, gender and basic demographic information will be obtained from items using the South African Census.[27] | – | – |
| Anthropometric measurements | Height and weight will be obtained to assess physical development. | – | – |
| Household food security | Study-specific items adapted from the household food insecurity access scale (10 items), which measures food security in the previous 30 days. | Eg, Did you worry that your household would not have enough food? Response: never/rarely/sometimes/often | Marlow et al (South Africa)[28] Knueppel et al (Tanzania)[29] Kabalo et al (Ethiopia)[30] |
| Socioeconomic status | Do you have running water? Do you have electricity? Do you have a flush toilet? | Eg, Do you have running water at your house? Response: yes/no | Du Toit et al (South Africa)[31] |
| | Socioeconomic status will also be measured by measuring access to the top eight socially perceived necessities for children, as identified by the Centre for South African Social Policy in the 'Indicators of poverty and social exclusion project'[32] and endorsed by over 80% of the South African population in a nationally representative survey (the South African Social Attitudes Survey 2006).[33] These include items such as 'enough clothes to keep you warm and dry'. | Please tick all of the things that you can afford at home.... Response: three meals a day, school fees | Cluver et al (South Africa)[34] Steventon Roberts et al (South Africa)[35] |
| | Cash grant receipt. | Eg, Are you or anyone in your household receiving any grants? | – |
| | Household employment. | Eg, How many people in your household are employed? | – |
| Household information | Study-specific questions on household arrangements and living circumstances. | Eg, What type of house do you live in? Response: house made of bricks on a separate yard/block of flats | – |
| **Health** | | | |
| COVID-19 | Study-specific questions on experience of COVID-19 including caregiver death, COVID-19 infection within the household, experience of lockdown and presence of long COVID. | Eg, Were you or anyone in your household infected with COVID-19? Response: yes/no | – |
| HIV | Study-specific questions on household HIV status, testing, status and treatment. | Eg, Is anyone in your household living with HIV? Response: yes/no/do not know/decline to answer | – |

Continued

**Table 1** Continued

| Domain | Measures | Example items and response categories | Examples of previous use and/or validation within LMIC |
|---|---|---|---|
| Overall health status | An adapted subscale from the International Classification of Functioning Disability and Health (3 items) will be used to assess self-reported health within the previous 12 months.[36] | Eg, In the last 12 months, did you have difficulty remembering things or following a story or conversation? Response: no, no difficulty at all/yes, some difficulty/yes, a lot of difficulty/ cannot do it at all | – |
| Sexual and reproductive health | Study-specific questions adapted from the National Survey and risk behaviour among South Africans[37] will be used to explore sexual and reproductive health. Items focus on sexual activity, age of sexual debut, contraception use, transactional sex, substance use during sexual activity, pregnancy and sexual abuse. | Eg, Are you currently in a sexual and/ or romantic relationship or have you ever been in a sexual and/or romantic relationship? Response: yes/no | – |
| Health service utilisation | Study-specific questions regarding what health services have been accessed by the participant in the previous 12 months. | Eg, Which of the following health services have you accessed in the last 12 months? | – |
| **Mental health** | | | |
| Depressive symptoms | The Child Depression Inventory Short Form (10 items)[38] was used to assess depressive symptomology within the previous 2 weeks. | Eg, Select the statement which best describes how you have felt in the past 2 weeks. Response: nothing will ever work out for me/I am not sure if things will work out for me/things will work out for me OK | Suliman (South Africa)[39] Cluver et al (South Africa)[34] Sherr et al (South Africa, Malawi, Zambia)[40] Roberts et al (South Africa)[41] |
| | The Patient Health Questionnaire-9 (9 items)[42] was used to assess depressive symptomology within the previous 2 weeks. | Eg, Have you felt down, depressed, irritable or hopeless? Response: not at all/several days/more than half the days/nearly every day | Aggarwal et al (South Africa)[43] Mudra Rakshasa-Loots et al (South Africa)[44] Chibanda et al (Zimbabwe)[45] |
| Anxiety symptoms | The Generalised Anxiety Disorder-7 (7 items)[46] was used to assess anxiety symptoms in the previous 2 weeks. | Eg, Have you felt nervous, anxious or on edge? Response: not at all/several days/more than half the days/nearly every day | Kigozi (South Africa)[47] Adjorlolo (Ghana)[48] Roberts et al (South Africa)[41] Chibanda et al (Zimbabwe)[45] Mughal et al (LMIC)[49] |
| Post-traumatic stress symptoms | A single item informed by the Child PTSD checklist was used to explore post-traumatic stress symptoms among those children and adolescents who had lost a caregiver. | Eg, How stressed have you been since the death of your caregiver? Response: very/a little/no more than usual | Boyes et al (South Africa)[50] Roberts et al (South Africa)[41] |
| Suicidality and self-harm | Mini International Psychiatric Interview for Children and Adolescents Suicidality (5 items)[51] was used to assess suicidality and self-harm behaviours in the previous month. | Eg, In the past month, did you wish you were dead? Response: yes/no | Sheehan et al and Lecrubier et al (Global)[52] [53] Cluver et al (South Africa)[34] Roberts et al (South Africa)[41] |

Continued

**Table 1** Continued

| Domain | Measures | Example items and response categories | Examples of previous use and/or validation within LMIC |
|---|---|---|---|
| Self-esteem | The Rosenberg Self-Esteem Scale (10 items)[40] was used as an assessment of child and adolescent self-esteem. | Eg, I feel that I have a number of good qualities Response: strongly agree/agree/ disagree/strongly disagree | Sherr et al (South Africa, Malawi, Zambia)[40] |
| Access to mental health services | A single study-specific item to assess whether children and adolescents have access to mental health services. | Eg, Do you have access to mental health services? Response: yes, needed and accessed/ yes, but did not need so did not access/ no, needed but could not access/no, did not need and did not access | – |
| **Bereavement experience** | | | |
| Grief | The Core Bereavement Items—Grief subscale (five items)[54] will be used to assess grief intensity. | Eg, Do reminders of X such as photos, situations, music, places etc. cause you to feel longing for X? Response: a lot of the time/quite a bit of the time/a little bit of the time/never | Thurman et al (South Africa)[55] |
| **Parenting and caregiving experience** | | | |
| Parenting experience | The Alabama Parenting Questionnaire (16 items) will be used to identify experience of positive parenting, caregiver supervision, experience of caregiver discipline within the previous 2 months.[56] | Eg, Thinking about the person who looks after you the most, your parent or caregiver says that you have done something well… Response: never/rarely/sometimes/ often/always | Cluver et al (South Africa)[57] Elgar et al (Global)[56] |
| Experience of caregiving and responsibility for household tasks | The Young Carers Tasks & Outcomes Questionnaire (23 items)[58] was adapted to assess children and adolescents experience of having responsibility for household tasks. | Eg, In the past week, have you washed other people's clothes? Response: yes/no How many days a week do you do this? | Lane et al (South Africa)[59] |
| **Support interventions and provision** | | | |
| Organisational support | Study-specific questions on support from organisations, for example, support from social services, healthcare workers. | Eg, Do you have access to an organisation that provides you with any support services? Response: yes/no | – |
| Community provision | Study-specific questions on community provisions. | Eg, Are you a member of any youth and/or health organisations, political or activist groups? | – |
| **Education and school experiences** | | | |
| Education | Study-specific questions on school enrolment, being in the correct class for age, educational attainment, education aspirations and school dropout. | Eg, Are you currently enrolled in school? Response: yes/no | – |

Continued

**Table 1** Continued

| Domain | Measures | Example items and response categories | Examples of previous use and/or validation within LMIC |
|---|---|---|---|
| **Behavioural risk** | | | |
| Behavioural problems | The Child Behaviour Checklist: Rule-breaking subscale (17 items) was used to assess behavioural problems in the previous 6 months.[60] | Eg, I drink alcohol to have fun, without my caregivers knowing or approving Response: not true/somewhat or sometimes true/very true or often true | Sherr et al (South Africa, Malawi, Zambia)[40] |
| **Violence exposure, stigma and bullying** | | | |
| Violence exposure | Study-specific questions adapted from the UNICEF measures for national-level monitoring of orphans and vulnerable children[61] and the Parent-Child Conflict Tactics Scales were used to assess household, domestic and school violence. | Eg, How often has anyone in your family or living in your home or someone at school used a stick, belt or other hard item to hit you? Response: weekly/monthly/at least once a year/has happened but not in the last year/never | – |
| Community violence | The child exposure to community violence checklist[62] will be used to identify exposure of children and adolescents to community violence. | Eg, Have you ever been hot or attacked outside your home? Response: yes, more than a year ago/ yes, in the last year/never | Sherr et al (South Africa, Malawi, Zambia)[40] Steventon Roberts et al (South Africa)[63] |
| Stigma | Study-specific questions focusing on COVID-19-related stigma. | Eg, I have been treated badly because I lost a caregiver to COVID-19 Response: not at all/sometimes/all of the time | – |
| Bullying | The Social and Health Assessment Peer Victimisation Scale (12 items) was used to examine experiences of bullying in the last 6 months.[64] | Eg, In the past 6 months, other kids have called me names or swore at me… Response: not at all/once/2–3 times/4 or more times | Boyes and Cluver (South Africa)[65] Sherr et al (South Africa, Malawi, Zambia)[40] |

LMIC, low-income and middle-income countries; PTSD, post-traumatic stress disorder.

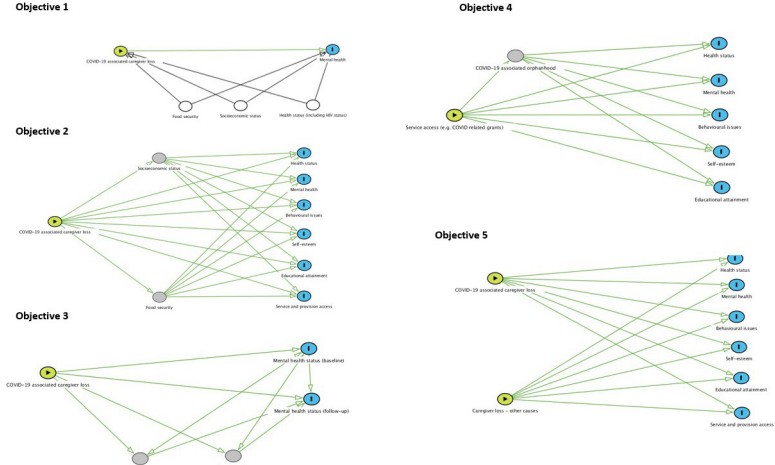

**Figure 1** Directed acyclic graphs illustrative of the OCAY study objectives (for illustrative purposes only). Objective 1: to determine if mental health outcomes (positive/negative) differ between children and adolescents who have experienced a COVID-19-associated caregiver death, and children and adolescents who have not experienced a COVID-19-associated caregiver death; objective 2: to explore the impact of COVID-19-associated caregiver death on health and well-being outcomes of children, and service use (comparing those with and without a caregiver loss, and exploring impacts in relation to sociodemographic factors, eg, experience of poverty); objective 3: to explore how mental health outcomes (positive/negative) among children and adolescents who have experienced orphanhood and caregiver loss change over time (using follow-up data); objective 4: to examine associations between access to services (such as COVID-related grants, bereavement referrals) and outcomes for children and adolescents; objective 5: to compare the health and well-being status of children and adolescents who have lost a caregiver due to COVID-19 with children and adolescents who have lost a caregiver due to other causes (eg, HIV/AIDS).

retention rates within previous studies among similar populations (eg, the Child Community Care study—89%, the Mzantsi Wakho study—95%).

### Data analyses

Descriptive statistics will be used to explore data yielded from the questionnaire to explore a range of outcomes for children and adolescents at each of the study time-points. Explorations will be based on study objectives. For illustrative purposes and to guide possible analyses, figure 1 presents directed acyclic graphs for each of the study objectives. A combination of analytical techniques will be used to explore different research questions inclusive of descriptive statistics (eg, t-tests and $\chi^2$ tests), analyses of variance, multilevel regression modelling to account for potential confounders and repeat measures and structural equation modelling (path analyses) with multiple outcomes to explore mediating and moderating factors in the relationship between predictor and outcome variables (identified a priori). Logistic regression modelling will be used to assess binary outcomes and linear regression modelling will be used to assess continuous outcomes. If possible, the above analytical techniques will also be used to explore additional data from existing sources to compare the health and well-being outcomes of children and adolescents experiencing COVID-19-associated loss with other groups, for example, children and adolescents experiencing caregiver loss due to HIV/AIDS.

## ETHICS AND DISSEMINATION

This study has received ethical approval from the Health Research Ethics Committee at Stellenbosch University (N22/04/040). This research is designed with child protection principles in mind, and the impact of the questionnaire is a foremost concern. The study will follow ethical procedures used by the researchers in other similar studies working with similar at-risk groups. The questionnaire will be designed to be child-friendly. All consent procedures will be conducted prior the commencement of the assessment. Given the young age of some participants and the sensitive nature of the project, all screening and consent processes will be conducted in the homes of the prospective participants. Hopefully, conducting screening and consent procedures in the comfort of the participants' homes will build trust and enhance rapport with both the participants and the caregivers. Caregivers will be welcomed to accompany their child to the interview, however, if this is not possible, written consent will be obtained prior to the scheduled appointment and permission will be sought to transport the child to and from the research centre or interview venue. Data collectors will explain the interview and assessment procedures to each child, and they will sign an assent form. Caregivers will sign a consent form to provide permission for their child to participate in the study. All participants will be assured that their decision to participate or not to participate in the study is completely voluntary and will not affect them in any way. All data will be anonymised prior to data analysis.

Links will be made with various NGOs and service providers prior to the commencement of the interviews to ensure efficient referral pathways. The research team anticipates a high number of referrals. As such, before the commencement of the interviews, various service providers were approached to ensure that the data collection team had a comprehensive referral directory. The referral directory has service providers in the fields of grief and loss, mental health, domestic violence, bullying and food insecurity. In addition, the participants will be provided with transport for their first appointment following referral.

Results will be disseminated via academic and policy publications, as well as national and international presentations inclusive of high-level meetings with technical experts. Findings will also be disseminated at a community level (eg, through specially convened community stakeholder meetings, NGOs, community leaders or local radio).

This study aims to provide early data on the experiences of children and adolescents aged 9–18 years following the death of a parent or caregiver as a consequence of the COVID-19 pandemic in South Africa. The study will provide insight into their needs, experiences and interventions that were provided, with an indication of gaps in provision, unmet needs and areas where future support could be targeted to optimise well-being in these children. Such data additionally aim to support planning for future pandemic responses. The use of a comparison group within this study will additionally allow for the explorations regarding the possible differences in experiences among children and adolescents who had experienced caregiver loss compared with children and adolescents who had not experienced caregiver loss (eg, experiences of lockdown). Similarly, these data will allow for the exploration of cumulative risk among children and adolescents living in contexts of adversity (eg, experiencing COVID-19-associated loss, poverty and living in HIV-affected communities), and help to identify the potential specific challenges related to the experience of caregiver loss in the wake of the COVID-19 pandemic.

Our research will generate evidence to inform further research, policy and programming for at-risk orphans and vulnerable children, including mental health screening and modifiable risk and protective factors for health and well-being among this group. Enhanced knowledge about what works for orphans and vulnerable children can be used to inform child health policies on a broad scale. We intend for our study to have wide applicability and have a clear path to dissemination. It is imperative to protect children and adolescents from the direct and indirect harms of the COVID-19 pandemic, both in the present and the long term. Data from this study provide a timely assessment of health and well-being in relation to caregiver loss with the aim of informing future responses to promote the potential of children and adolescents in South Africa and beyond.

**Author affiliations**
[1]Department of Social Policy and Intervention, University of Oxford, Oxford, UK
[2]Institute for Global Health, University College London, London, UK
[3]Institute for Life Course Health Reseach, Department of Global Health, Stellenbosch Univeristy, Stellenbosch, South Africa
[4]School of Nursing & Midwifrey, Queens University, Belfast, UK
[5]Department of Psychiatry & Mental Health, Univeristy of Cape Town, Cape Town, South Africa
[6]Amsterdam Institute for Social Science Research, Faculty of Social & Behavioural Sciences, Univeristy of Amsterdam, Amsterdam, Netherlands

**Contributors**  LS, LDC, MT, SS conceptualised the research study. All authors supported the development of the study, conceptualised research question, questionnaires and methodologies. KJSR, SDT, TM and CAL drafted the initial manuscript and led subsequent drafts. All authors read and provided feedback on manuscript iterations. All authors approved the final manuscript.

**Funding**  UKRI Global Challenges Research Fund (GCRF) Accelerating Achievement for Africa's Adolescents (Accelerate) Hub (Grant Ref: ES/S008101/1); Oak Foundation/GCRF 'Accelerating Violence Prevention in Africa' (OFIL-20-057); UNICEF ESARO; Wellspring Philanthropic Fund (grant no. 16204).

**Competing interests**  None declared.

**Patient and public involvement**  Patients and/or the public were not involved in the design, or conduct, or reporting, or dissemination plans of this research.

**Patient consent for publication**  Consent obtained from parent(s)/guardian(s).

**Provenance and peer review**  Not commissioned; externally peer reviewed.

**ORCID iDs**
Kathryn J Steventon Roberts http://orcid.org/0000-0003-1200-5362
Mark Tomlinson http://orcid.org/0000-0001-5846-3444
Lorraine Sherr http://orcid.org/0000-0002-5902-8011

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
