## [Reviewer comments · BMJ Open]

ARTICLE DETAILS

TITLE (PROVISIONAL)	Protocol for the OCAY study: A cohort study of orphanhood and caregiver loss in the Covid-19 era to explore the impact on children and adolescents
AUTHORS	Steventon Roberts, Kathryn; Du Toit, Stefani; Mawoyo, Tatenda; Tomlinson, Mark; Cluver, Lucie; Skeen, Sarah; Laurenzi, Christina; Sherr, Lorraine

VERSION 1 – REVIEW

REVIEWER	Spencer, Nick University of Warwick, School of Health and Social Studies
REVIEW RETURNED	09-Jan-2023

GENERAL COMMENTS	This is a potentially important study and is consistent with the following recommendation we made in our recent BMJPO editorial: The modelling studies have exposed and highlighted a major adverse consequence of the pandemic for children globally. However, further research is needed to identify the nature and extent of this 'hidden pandemic' and to inform policy to ensure children are protected against exploitation and abuse. Empirical research based on existing cohort studies and national survey data linked to mortality data is needed to fulfil these aims (BMJ Paediatrics Open 2022;6:e001604. doi:10.1136/ bmjpo-2022-001604 – p.2) Judging by the extensive citations of previous publications by the authors, this is an experienced research team that is likely to be able to complete the study. I have some comments and suggestions for clarification of the study that hopefully will strengthen the proposal. 1. Consistent with BMJ Open guidelines for study protocols, the authors must provide dates for the proposed start and finish of the study2. Study objectives: The objectives listed (lines 35-45) are appropriate to the study although I would suggest including an equity objective as the poorest, most marginalized groups are likely to be most vulnerable to the affects of orphanhood/care giver loss on mental and physical as well as economic wellbeing. Although the listed objectives are appropriate, I suggest adding research hypotheses and specific research questions against which the results can be assessed. Data will be collected in the study in 2 waves, 18 months apart; however, the objectives do not include any reference to changes over time and the authors should address this.3. Participant recruitment: More information is needed on recruitment from existing studies. How were these recruited and is the use of these participants likely
--

	to bias the sample i.e. are the populations from which existing studies were recruited significantly different to the population in Khayelitsha? Are the affected and non-affected children from the same population/s? 4. Study measures: Table 1 gives a comprehensive range of measures, many of which are validated, that will form part of the study specific inventory the authors are constructing. Even if it is still in draft form, I think it would be informative to include it in supplementary material so the reader has a clear view of the format of the inventory. It is not possible to tell from the measures outlined in the table if changes in mental health and wellbeing that may have occurred during the pandemic independent of the loss of parents/carers. Given the evidence for increased mental health problems generally among children and young people in the pandemic, I think the authors should ensure that questions are included on this issue. 5. Data collection procedures: No mention is made in the procedures to the second wave interviews and any anticipated differences with the first wave. Attrition may be a significant problem and the authors should provide information on proposed measures to ensure participant retention. The authors should also indicate whether the inventory will be modified for the second wave of data collection. 6. Pilot: The authors plan to pilot the study measures (line 46) although they give no details of numbers and potential participants. A brief outline of the pilot should be included. 7. Data analyses: The authors state “A combination of analytical techniques will be utilised to explore different research questions” (lines36-7). As pointed out above, no research questions are currently included in the protocol so it is not possible to decide if the proposed combination of analytical techniques is appropriate. Inclusion of research questions would enable full assessment of the proposed techniques and the addition of a Directed Acyclic Graph (DAG) would assist the reader in visualising the key elements of the analysis. Summary: An important study which will contribute to a more detailed understanding of the impact of orphanhood and care giver loss during the pandemic on children and young people; however, I think the protocol will be strengthened by attention to the issues raised above.
--	--

REVIEWER	Clavenna, Antonio Istituto di Ricerche Farmacologiche Mario Negri, Laboratory for Mother and Child Health, Department of Public Health, Department of Public Health
-----------------	---

REVIEW RETURNED	13-Mar-2023
-------------

GENERAL COMMENTS	In my opinion, a few clarifications may be helpful. 1. Which is the time period of the study? In my opinion the interval between the loss of the parent/caregiver (or the lockdown experience) and the baseline interview has an influence on the results of the evaluation, and it seems to me that you have to consider this variable.
--

	2. If possible, it could be important to compare the well-being and the health status of children who loss a parent or a caregiver for COVID-19 also with a group who loss caregiver for other causes (not related to COVID-19).
--	--

VERSION 1 – AUTHOR RESPONSE

Reviewer: 1

Prof. Nick Spencer, University of Warwick Comments to the Author:

This is a potentially important study and is consistent with the following recommendation we made in our recent BMJPO editorial:

The modelling studies have exposed and highlighted a major adverse consequence of the pandemic for children globally. However, further research is needed to identify the nature and extent of this 'hidden pandemic' and to inform policy to ensure children are protected against exploitation and abuse. Empirical research based on existing cohort studies and national survey data linked to mortality data is needed to fulfil these aims (BMJ Paediatrics Open 2022;6:e001604. doi:10.1136/bmjpo-2022-001604 – p.2) Judging by the extensive citations of previous publications by the authors, this is an experienced research team that is likely to be able to complete the study. I have some comments and suggestions for clarification of the study that hopefully will strengthen the proposal.

Thank you for your comments. Please see responses below.

1. Consistent with BMJ Open guidelines for study protocols, the authors must provide dates for the proposed start and finish of the study

Scheduled dates for data collection have now been added to the protocol under the study measures section.

2. Study objectives:

The objectives listed (lines 35-45) are appropriate to the study although I would suggest including an equity objective as the poorest, most marginalized groups are likely to be most vulnerable to the affects of orphanhood/care giver loss on mental and physical as well as economic wellbeing.

Although the listed objectives are appropriate, I suggest adding research hypotheses and specific research questions against which the results can be assessed.

Data will be collected in the study in 2 waves, 18 months apart; however, the objectives do not include any reference to changes over time and the authors should address this.

Thank you for the suggestion. Given the setting of this study (Khayelitsha), the focus will predominantly be on those children and adolescents which are likely to be marginalised and experiencing poverty. Additional detail has now been added to the objectives relating to experience of poverty.

Objective 3 includes reference to measuring mental health changes over time however, an additional note has been added to this section for clarity.

A statement relating to the study hypotheses has now been added to this section. The decision was made not to include research questions as these were repetitive of the objectives already outlined.

3. Participant recruitment:

More information is needed on recruitment from existing studies. How were these recruited and is the use of these participants likely to bias the sample i.e. are the populations from which existing studies were recruited significantly different to the population in Khayelitsha?

Are the affected and non-affected children from the same population/s?

Further information relating to recruitment from existing studies has now been added to the participant recruitment section. These studies utilise populations with similar sociodemographic characteristics to the population of interest and participants have previously given consent to be contacted for future studies. This was used as a starting point for data collection and was included as one of multiple strategies for recruitment. Data collection has now started and only a very small proportion of participants were recruited via this method, limiting potential bias.

4. Study measures:

Table 1 gives a comprehensive range of measures, many of which are validated, that will form part of the study specific inventory the authors are constructing. Even if it is still in draft form, I think it would be informative to include it in supplementary material so the reader has a clear view of the format of the inventory.

Due to the length of the questionnaire 50+ pages, the decision was made to not include the full questionnaire in supplementary material. An overview of measures is provided in Table 1.

It is not possible to tell from the measures outlined in the table if changes in mental health and wellbeing that may have occurred during the pandemic independent of the loss of parents/carers. Given the evidence for increased mental health problems generally among children and young people in the pandemic, I think the authors should ensure that questions are included on this issue.

Thank you for your suggestion. Mental health and wellbeing measures are included within the study questionnaire irrespective of child orphanhood status. These measures are obtained independent of loss status and analytical methods will be used to explore mental health and wellbeing in relation to loss and independent of loss (e.g., by controlling for loss status). Mental health and wellbeing will be explored in both the orphanhood affected group and the non-affected group. Further detail has been added to the study measures section for clarity.

5. Data collection procedures:

No mention is made in the procedures to the second wave interviews and any anticipated differences with the first wave. Attrition may be a significant problem and the authors should provide information on proposed measures to ensure participant retention. The authors should also indicate whether the inventory will be modified for the second wave of data collection.

Further detail relating to follow up data collection has now been added to the data collection procedures section. Detail relating amendments to the follow-up questionnaire have now been added to the study measures section of the manuscript.

6. Pilot:

The authors plan to pilot the study measures (line 46) although they give no details of numbers and potential participants. A brief outline of the pilot should be included.

Detail relating to piloting has now been added to the section relating to study measures.

“ Piloting will be undertaken with 10-15 participants who have been identified as eligible for the study within the Khayelitsha area”

7. Data analyses:

The authors state “A combination of analytical techniques will be utilised to explore different research questions” (lines36-7). As pointed out above, no research questions are currently included in the protocol so it is not possible to decide if the proposed combination of analytical techniques is appropriate. Inclusion of research questions would enable full assessment of the proposed techniques

and the addition of a Directed Acyclic Graph (DAG) would assist the reader in visualising the key elements of the analysis.

Thank you for your suggestion. The research questions for the study are very similar to the objectives and it was deemed repetitive to include both. However, to guide the reader, illustrative DAGS relating to each objective have now been added within Figure 1.

Summary:

An important study which will contribute to a more detailed understanding of the impact of orphanhood and care giver loss during the pandemic on children and young people; however, I think the protocol will be strengthened by attention to the issues raised above.

Thank you for your comments.

Reviewer: 2

Dr. Antonio Clavenna, Istituto di Ricerche Farmacologiche Mario Negri Comments to the Author:

In my opinion, a few clarifications may be helpful.

Thank you for your comments. Please see responses below.

1. Which is the time period of the study? In my opinion the interval between the loss of the parent/caregiver (or the lockdown experience) and the baseline interview has an influence on the results of the evaluation, and it seems to me that you have to consider this variable.

Scheduled study dates have now been added to the study measures section of the manuscript. We agree that time since caregiver loss should be accounted for within analyses and this was planned. Further detail has now been added to the study measures section to make this intention clear.

2. If possible, it could be important to compare the well-being and the health status of children who loss a parent or a caregiver for COVID-19 also with a group who loss caregiver for other causes (not related to COVID-19).

Thank you for this suggestion. This has now been added as an additional objective. Where possible this will be explored in the current OCAY dataset, however further detail has now been added to the data analyses section of the manuscript relating to the possible use of existing data to compare the health and wellbeing outcomes of children and adolescents experiencing COVID-19 associated caregiver loss, and children and adolescent who have lost a caregiver due to other sources e.g., HIV/AIDS.

VERSION 2 – REVIEW

REVIEWER	Spencer, Nick University of Warwick, School of Health and Social Studies
REVIEW RETURNED	04-May-2023
GENERAL COMMENTS	The authors have responded appropriately to my comments and I now think the protocol should be accepted for publication.